# Associations between Muscle Strength, Physical Performance and Cognitive Impairment with Fear of Falling among Older Adults Aged ≥ 60 Years: A Cross-Sectional Study

**DOI:** 10.3390/ijerph191710504

**Published:** 2022-08-23

**Authors:** Antonio Orihuela-Espejo, Francisco Álvarez-Salvago, Antonio Martínez-Amat, Carmen Boquete-Pumar, Manuel De Diego-Moreno, Manuel García-Sillero, Agustín Aibar-Almazán, José Daniel Jiménez-García

**Affiliations:** 1Faculty of Sport Sciences, EADE-University of Wales, Trinity Saint David, 29018 Málaga, Spain; 2Department of Health Sciences, Faculty of Health Sciences, University of Jaén, 23071 Jaén, Spain; 3Department of Physiotherapy, Faculty of Health Sciences, European University of Valencia, 46112 Valencia, Spain; 4Laboratory FiveStars, 29018 Málaga, Spain

**Keywords:** muscle strength, cognitive impairment, fear of falling, older adults

## Abstract

(1) Background: Fear of falling has become a significant health problem in older adults and is already considered as important as falling because of its long-term detrimental effects on older adults’ physical and psychosocial functioning. The aim of this study was to analyze the associations between both upper and lower limb strength, gait parameters and cognitive impairment with fear of falling in older adults. (2) Methods: A cross-sectional study involving 115 older-adult participants was used to assess the impact of both upper (Handgrip dynamometer, TKK 5401 Grip-D, Takey, Tokyo, Japan) and lower limb strength (Chair stand test), gait parameters (OptoGait-System Bolzano, Bolzano, Italy) and cognitive impairment (COWAT word association test) with fear of falling in older adults (Falls Efficacy Scale-International FES-I). (3) Results: Multivariate linear regression analysis showed several independent associations with the fear of falling. A higher time to perform the Chair Stand test was associated with higher scores in FES-I (R^2^ = 0.231), while a lower score in both Semantic Fluency (S COWA) and Phonologic Fluency (P COWA) was associated with a decreased score in FES-I (R^2^ = 0.052 and 0.035). (4) Conclusions: Both higher step and stride length (OptoGait), lower body strength (Chair test) and both poorer semantic (S COWA) and phonologic (P COWA) fluency were all associated with fear of falling.

## 1. Introduction

The aging of the global population and its ensuing social and economic issues have emerged as one of the most pervasive challenges in recent years [1]. The World Health Organization predicts that there are 728 million people over the age of 65 worldwide as of right now, with 1 billion in 2030 and 2.5 billion in 2100 [2]. As the proportion of older adults in society increases, fear of falling has become a significant health problem in older adults and is already considered as important as falling [3]. In this sense, fear of falling is defined as ‘a lasting concern on falling that can lead to an individual avoiding activities that he/she remains capable of performing’, indicating that it is associated with prior fall experience [4].

According to the different measurements used in previous studies, the reported prevalence of fear of falling ranges from 21 to 85% among community-dwelling older adults who have previously fallen, and between 33 and 46% in those who have not fallen [4,5,6,7]. Numerous studies have shown that it has long-term detrimental effects on older adults’ physical and psychosocial functioning, including an increased risk of falling, activity restriction or avoidance, a lack of independence and confidence, poorer mental health, depression and a decreased quality of life [3]. When it comes to psychological impact, fear of falling might even be worse than actually falling [8]. Thus, the detrimental effects on daily living and quality of life in older adults, along with associated health care costs and use of resources, are of serious concern not only to older adults themselves but also to caregivers and health and social services [9].

Because of the negative impact of fear of falling and its consequences, several studies have already remarked that the risk factors of fear of falling include a previous history of falls, female gender, older age, lower levels of economic resources, use of seven or more medications, hearing impairment, functional dependence, diminished gait speed, self-reported poor health, poorer both upper and lower extremity muscle strength, cognitive decline and symptoms of anxiety and depression [3,10,11,12,13,14,15,16,17,18,19]. However, it is frequently challenging to ascertain the true influence of risk factors because not all research evaluates both physical and psychological variables in the same population as potential risk factors and also by the frequency with which different tools to quantify fear of falling are used. Hence, the interpretation of findings is often limited and ambiguous, and the findings cannot be generalized. In this regard, considering that fear of falling is multifactorial, future studies should therefore focus on both physiological and psychological factors in the same adult population, which would allow not only to provide guidance when creating rehabilitation programs for older adults to overcome the fear of falling, but also to have a positive direct impact on the quality of life of this population.

Thus, the objective of this study was to analyze the associations between both upper and lower limb strength, gait parameters and cognitive impairment with fear of falling in older adults.

## 2. Materials and Methods

### 2.1. Study Design and Participants

This was a cross-sectional descriptive and analytical study that involved a total of 115 older adults over 60 years of age, who were randomly selected through the department of Social Affairs of the City Council of Malaga and the City Council of Pizarra (Malaga) (Figure 1). The inclusion criteria for this study were: adults over 60 years of age who are registered as residents of the municipalities of Malaga and Pizarra. Exclusion criteria were: having central or peripheral neurological alterations, rheumatological diseases, serious somatic or psychiatric diseases, severe cognitive impairment, pacemakers or prostheses and any type of alteration or pathology that could alter balance and functional activity (such as auditory or vestibular alterations). This study was approved by the Ethics Committee of the University of Jaén with protocol code DIC.17/5.TES (approved on 19 February 2018) and was carried out taking into account the guidelines of the Declaration of Helsinki. Before the study was carried out, all participants signed their informed consent.

When a selected person could not be included in the sample, either for not having access to it, for not showing predisposition or for not meeting the inclusion criteria, they were replaced by the next person on the list, with the aim of maintaining the reference sample size (determination of the sample size was conducted using the G*Power software (Version 3.1.9.2; Heinrich-Heine-Universität Düsseldorf, Düsseldorf, Germany)).

Estimating a 30% incidence of falls (at least once a year) in the study population, the study sample size was calculated using the formula for estimating proportion, where the expected rate was 0.3, with α = 0.05 and precision of ±0.06. By adding 25% to counteract recording errors, exclusions or people who did not wish to participate, a necessary sample size of 179 subjects was obtained.

Once the sample was obtained and recruitment was completed, participants and their families received talks to understand the program and its procedure.

### 2.2. Study Outcomes

#### 2.2.1. Sociodemographic and Anthropometric Data

Data such as full name, age, date of birth, marital status, educational level and place of residence were recorded. Anthropometric data (weight, height, waist–hip index and BMI) and lifestyle habits related to the practice of physical activity, type and frequency, smoking and falls in the last year were also included.

Furthermore, to measure the abdominal perimeter (waist), a 1.5 m flexible tape (Lufkin, W606PM, Boston, MD, USA) was used, taking as reference the equidistant point between the last rib and the iliac crest. Two measurements with the participant in a standing position were taken. 

#### 2.2.2. Fear of Falling

This aspect, together with confidence in performing daily activities, was evaluated using the Falls Efficacy Scale-International (FES-I) in its Spanish version [20]. This is a scale composed of 16 items that assess the risk of falling during the performance of basic activities of daily living. The score for each item was indicated by the respondent, ranging from 1 (no fear/concern about falling) to 4 (severe fear/concern about falling). The result is a total score between 16 and 64 points, with a higher score correlating with greater concern about falling. The Spanish version of the FES-I was validated in a postmenopausal population [21].

#### 2.2.3. Lower Limb Strength

To assess lower limb strength, the Chair Stand Test [22] was used, consisting of standing up and sitting on a chair five times as fast as possible, keeping the arms crossed at chest height. Depending on the time invested, the score for this test is: 4 points = less than 11.19 s; 3 points = between 11.20–13.69 s; 2 points = between 13.70–16.69 s; 1 point = between 16.7–59 s; 0 points = more than 60 s or unable.

#### 2.2.4. Upper Extremity Muscle Strength

An assessment of manual grip was performed using a manual hydraulic dynamometer (TKK 5401 Grip-D, Takey, Tokyo, Japan), commonly used for the assessment of muscle strength in this type of study.

#### 2.2.5. Gait Parameters

Different researchers have studied gait and stride length, speed, acceleration and duration as predictors of fall risk, dependency, institutionalization and mortality. [23,24,25]. In this task, the use of different technological instruments was accompanied by more traditional assessment. Among the sensory elements used, optoelectronic assessment systems have become widespread due to their reliability and cost.

In our study, an optoelectronic measuring system (OptoGait-System Bolzano, Bolzano, Italy) was used to evaluate gait parameters. This system is composed of a transmitting optical bar (with 96 infrared LED lamps) and a receiving one, which allows us to detect eventual interruptions and their duration, and, therefore, to obtain data related to gait, its length, speed, acceleration, flight and contact times, with high precision and reliability [26].

The participants walked for 2 min on a treadmill to adapt to the test, which has a duration of 30 s, walking at a constant speed of 3.5 km/h, providing sufficient distance to allow acceleration and deceleration from the resting state.

#### 2.2.6. Semantic and Phonologic Fluency

Semantic and phonologic fluency were assessed using the COWAT word association test [27]. This is a verbal fluency test that measures the spontaneous production of words that belong to the same category or that begin with a designated letter. The subject must produce orally, for 60 s, as many words as possible, beginning with a given letter of the alphabet. Proper names or derivations of the same word are not allowed.

The COWAT is a widely used procedure to assess verbal fluency, as it is a sensitive indicator of brain dysfunction. The administration of verbal fluency tasks is recognized as an important component in the comprehensive assessment of neuropsychological functioning [28,29].

### 2.3. Data Collection

Each evaluation comprised a personal interview, as well as the application of the assessment tests and was conducted by a previously trained evaluator.

### 2.4. Sample Size Calculation

For sample size calculation, at least 20 subjects per event were required in the multivariate linear regression model [30]. Thirteen possible events were used: lower limb strength score, upper extremity muscle strength score, step length, double support, step time, stride length, gait speed, acceleration and distance as gait parameters, semantic and phonologic fluency total scores, as well as age, sex, BMI and education status. Therefore, 110 participants were required for this analysis. The final number of participants was 115.

### 2.5. Statistical Analysis

Continuous variables were summarized as means and standard deviations, and categorical variables as percentages and frequencies. The Kolmogorov–Smirnov test was employed to evaluate the normal distribution of all variables. A bivariate correlation analysis was applied to evaluate the possible individual ways in which independent variables such as lower limb strength, upper extremity muscle strength, gait parameters and semantic and phonologic fluency, as well as other covariates such as BMI, sex, age and educational attainment level, are associated with fear of falling. In order to examine possible independent associations between study variables, both a multivariate linear regression model and a step-by-step method were employed to introduce variables into the model. The risk of falls was registered as a dependent variable in separate models (significant in bivariate correlation “*p* < 0.05”) and was incorporated in the multivariate linear regression. Adjusted R^2^ was utilized to calculate the effect size coefficient of multiple determination in linear models. If <0.02, R^2^ was deemed to be negligible, medium if between 0.02 and 0.15 and large if >0.35. A confidence level of 95% was used (*p* < 0.05). The SPSS statistical package for Windows (SPSS Inc., Chicago, IL, USA) was employed for data analysis.

## 3. Results

Table 1 display descriptive data of the participants. A total of 115 older adults (70.33 ± 8.16 years) took part in the present study. Most of the participants pursued at least primary education (75.66%), and the mean BMI was 29.34 ± 5.03 kg/m^2^. The descriptive data of the variables analyzed in this study and gait analysis of the descriptive data of the variables analyzed showed how gait speed was 1.11 ± 1.46 m/s, while distance was 3704.37 ± 6596.85. As for strength levels, handgrip strength was 23.24 ± 8.51 kg, while the time to perform the chair stand test was 14.32 ± 4.92 s. Regarding cognitive impairment, the number of words for semantic fluency was 28.35 ± 10.83 w, while the number of words for phonologic fluency was 32.07 ± 12.08 w. Considering the dependent variable, the score for FES-I was 24.68 ± 8.09.

The bivariate analysis (Table 2) showed that the dependent variable of the present work, fear of falling, was significantly positively correlated with both Step and Stride Length. Additionally, higher scores in FES-I were also correlated with an increased time to realize the chair stand test. Regarding semantic and phonologic fluency, fear of falling was significantly negatively correlated with both of them. Fear of falling was not correlated with age, sex and level of education in our participants.

Multivariate linear regression analysis (Table 3) showed several independent associations with the fear of falling. A higher time to perform the Chair Stand test was associated with higher scores in FES-I (R^2^ = 0.231), while a lower score in both Semantic Fluency and Phonologic Fluency was associated with a decreased score in FES-I (R^2^ = 0.052 and 0.035).

## 4. Discussion

The purpose of this study was to analyze the associations between both upper and lower limb strength, gait parameters and cognitive impairment in older adults with a fear of falling. The main findings of the study, which involved 115 older adults aged >60 years, point to a significant positive association between both higher step and stride length (OptoGait) and lower body strength (Chair test) with the fear of falling (*p* < 0.05). In contrast, significant negative associations were also observed between both poorer semantic (S COWA) and phonologic (P COWA) fluency with the fear of falling (*p* < 0.05). This suggests that not only physical but also cognitive features may play a role in predicting a higher fear of falling, which, in turn, could also be translated into a profound and largely detrimental effect on daily activities and into a worsening of these patient’s quality of life, as evidenced by earlier research [31,32,33]. 

Although earlier research previously looked at a probable link between some of these risk factors and the likelihood and apprehension of falling [11,12,13,14,15,16,17,18,19], to the best of our knowledge, this is one of the few studies to explore the impact of upper and lower limb strength, gait speed and cognitive impairment in the same sample. As a result, in addition to adding to the current research, our paper provides a broad understanding of how these characteristics may have a detrimental impact on the fear of falling and, therefore, on the quality of life of older adults. Both our step and stride length results show significant positive associations (*p* < 0.05) between both types of length and higher fear of falling in older adults. This is in agreement with the fact that longer step lengths and slower gait speeds have been shown to be directly associated with increased fear of falling [34,35,36] and are correlated with lower scores on clinical balance scales [37]. However, this seems to contradict the results found in previous research highlighting that lower speed is associated with greater stability [38]. Therefore, a possible explanation for this paradox in our results could be that older adults do not walk slowly enough in relation to their maximum walking velocity, resulting in a lower safety factor during normal locomotion, which can explain their less safe locomotion (i.e., older adults are more prone to falls during walking than young adults) [39]. Notwithstanding, we expected to find significant negative associations between both handgrip strength (measured through dynamometry) and gait speed with fear of falling, given that individuals with low grip strength and slower gait were previously linked to fragility, propensity to fall and higher apprehension to fall [40]. At this point, one possible explanation for why our study failed to find significant differences among our participants is that in most studies, the value of manual dynamometry is obtained by averaging three attempts in the dominant hand, whereas in our case, the total value for this variable was obtained by adding all the attempts between both hands (dominant and non-dominant), which may have influenced the results obtained. In terms of gait speed, we speculate that our findings may be due to the fact that gait analysis was conducted using a photoelectric cells system that assessed gait speed in conjunction with other gait parameters, whereas the majority of the studies use a single assessment tool, such as the Time up and Go Test (TUG), to reach conclusions.

In addition to the aforementioned aspects, our results suggest that there is a significant negative association between both poorer semantic (S COWA) and phonologic (P COWA) fluency with the fear of falling (*p* < 0.05). These findings corroborate prior research that has shown that people with decreased global cognitive function present higher fear of falling [41,42,43], although the specific cognitive domains that are most related to this apprehension are just beginning to be understood. In this regard, while some previous studies highlighted how impaired cognition was identified as a predictor of fear of falling [41,42,43], others did not show an association between cognitive impairment and fear of falling [44]. According to most of these authors, the differences between the results are mainly due to the great variability of instruments for assessing cognitive impairment among the different studies. For this reason, this is one of the few studies that explore and speculate how other different cognitive indicators, such as verbal fluency, may also play a role in raising the fear of falling in older adults. Therefore, future research should continue to take into account the variation in the assessment of cognitive impairment in order to better understand which mental abilities contribute the most to the fear of falling.

Finally, multivariate regression analysis showed associations between some predisposing factors with increased fear of falling. Longer periods of time to perform the chair test were associated with an increased FES-I score, while lower scores in both semantic (S COWA) and phonologic (P COWA) fluency were associated with an increase in the FES-I score.

This study has a number of limitations to be acknowledged. The cross-sectional design of the study does not allow causal relationships to be inferred, and the fact that this study was carried out among men and women equal to or older than 60 years from a specific geographical area means that any generalization of the results should be limited to people with similar characteristics.

Henceforth, health professionals and future studies should inquire more about which physical and cognitive characteristics enhance the likelihood and apprehension of falling in these patients, avoiding a negative impact on their quality of life.

## 5. Conclusions

The findings of this study suggest that, in older adults aged ≥60 years, both higher step and stride length (OptoGait), lower body strength (Chair test), and both poorer semantic (S COWA) and phonologic (P COWA) fluency were all associated with the fear of falling. Therefore, in order to manage and prevent the fear of falling and a possible risk of falling in this population, which could have a direct influence on their quality of life, these physical and cognitive characteristics should be taken into account.

## Figures and Tables

**Figure 1 ijerph-19-10504-f001:**
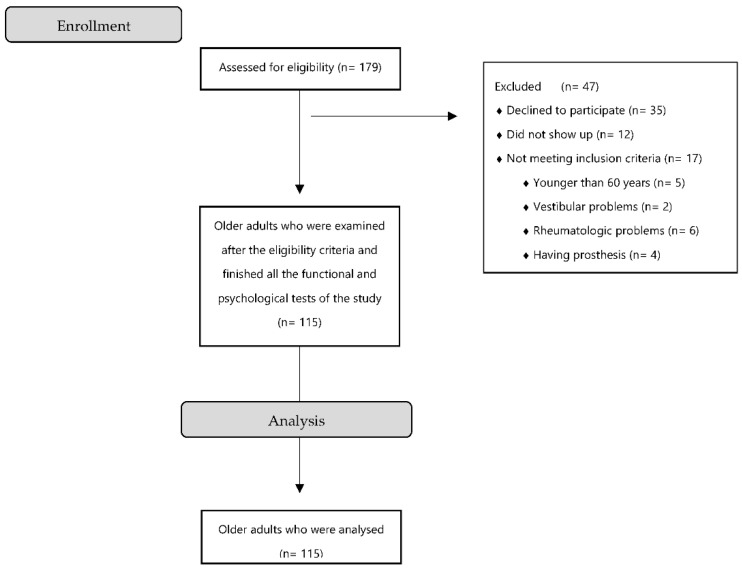
Flow diagram of study participants.

**Table 1 ijerph-19-10504-t001:** Descriptive data of the sample (*n* = 115).

Characteristics		Values Total = 115		Values	Men = 37	Values	Women = 78
Age (years)		70.33	8.16	71.94	8.59	69.56	7.89
BMI (kg/m^2^)		29.34	5.03	29.09	3.63	29.46	5.60
Educational status , *n* (%)	No formal education	16	13.92	4	10.82	12	15.38
	Primary education	71	61.74	28	75.67	43	55.12
	Secondary education	23	20	4	10.82	19	24.35
	University	5	4.34	1	2.69	4	5.14
S-COWA (words)		28.35	10.83	29.94	13.98	27.60	8.98
P-COWA (words)		32.07	12.08	30.75	13.97	32.70	11.11
Step length (cm)		56.29	12.83	56.24	14.24	56.32	12.19
Double support (s)		0.31	0.15	0.33	0.14	0.30	0.16
Step time (s)		0.47	0.21	0.48	0.18	0.47	0.21
Stride length (cm)		112.67	25.36	112.78	28.92	112.62	24.05
Gait speed (m/s)		1.11	1.46	0.89	0.21	1.21	1.76
Aceleration (s)		0.02	0.22	0.01	0.03	0.03	0.27
Distance (cm)		3704.37	6596.85	2952.67	525.82	4060.94	7993.75
Handgrip strength (kg)		23.24	8.51	31.28	9.07	19.43	4.78
Chair Stand (s)		14.32	4.92	13.92	5.40	14.53	4.70
FES-I score		24.68	8.09	24.64	8.95	24.70	7.72

BMI: Body Mass Index. S-COWA: Semantic Fluency. P-COWA: Phonologic fluency. FES: Falls Efficacy Scale International.

**Table 2 ijerph-19-10504-t002:** Pearson’s correlations between analyzed parameters in this study.

	FES Score (s)
S COWA	−0.229 ^1^
P COWA	−0.188 ^1^
Step length	0.222 ^1^
Double support	0.001
Step time	0.109
Stride length	0.222 ^1^
Gait speed	0.043
Acceleration	0.005
Distance	0.994
Handgrip strength	−0.075
Chair Stand	0.481 ^1^
Sex	0.003
Educational Status	−0.178
Age (years)	0.016
BMI (kg/m^2^)	0.143

S COWA: Semantic Fluency. P-COWA: Phonologic fluency. BMI: Body Mass Index. ^1^
*p* < 0.05.

**Table 3 ijerph-19-10504-t003:** Multivariate linear regression analyses for factors associated with fear of falling.

Variable		B	β	t	95% CI		*p*-Value
FES score (s)	S COWA	−0.180	−0.241	−2.459	−0.325	−0.035	0.015
	P COWA	−0.158	−0.235	−2.233	−0.297	−0.018	0.028
	Chair Stand	0.793	0.482	5.604	0.513	1.073	0.000

B: unstandardized coefficient. β: standardized coefficient. CI: confidence interval. S COWA: Semantic Fluency. P-COWA: Phonologic fluency.

## Data Availability

The data shown in this study are available upon request from the corresponding author. Data are not available to the public given the sensitive nature of the questions asked in this study and the necessary guarantees of privacy and confidentiality.

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
