# Peer review of "Associations between Muscle Strength, Physical Performance and Cognitive Impairment with Fear of Falling among Older Adults Aged ≥ 60 Years: A Cross-Sectional Study"

_ijerph, 2022, doi:10.3390/ijerph191710504_

Round 1

Reviewer 1 Report

The authors have adequately addressed all my comments.

Reviewer 2 Report

The authors have conducted a study to test the relationship between both upper and lower limb strength, gait parameters and cognitive impairment with fear of falling in elderly.

The study is well executed and presents interesting data. Below are some of the suggestion that the authors may want to consider.

Abstract: Aim is presented in the background section? please fix it.

Elderly is generally for someone above 75y. Please consider using "older adults" in place of elderly all over the manuscript.

Introduction: Meriam Nelson and Steve Ball have conducted many studies in older adults. The authors are recommended to consider reading their work and include the citations in the introduction and discussion as applicable. Examples below. The authors can look up other relevant work from these authors.

Morganti, C. M., Nelson, M. E., Fiatarone, M. A., Dallal, G. E., Economos, C. D., Crawford, B. M., & Evans, W. J. (1995). Strength improvements with 1 yr of progressive resistance training in older women. Medicine and science in sports and exercise27(6), 906-912.

Syed-Abdul, M. M., McClellan, C. L., Parks, E. J., & Ball, S. D. (2022). Effects of a resistance training community programme in older adults. Ageing & Society42(8), 1863-1878.

Methods: Please include the information related to ethics board, consenting, and approval numbers.

Fig 1 can be redesigned to follow CONSORT flow chart.

275-285 different font?

Discussion: Since there were more women in the group, how does the results change when only women are included in the analysis? This secondary analysis can be provided in the supplementary file if the findings were interesting.

Author Response

This manuscript is a resubmission of an earlier submission. The following is a list of the peer review reports and author responses from that submission.

Round 1

Reviewer 1 Report

The present work aimed to examine the associations between muscle strength (upper and lower body), gait parameters and cognitive impairment with the fear of falling among older adults. Unquestionably research on falls prevention in senescence is very important. The text in the manuscript is in general well written and easy to follow. However, I have two main concerns:

        1. The authors examined the association between fear of falling, which is a psychological characteristic, and muscle strength and gait parameters, which are physiological/biomechanical/functional characteristics as well cognitive impairment, which is again rather a psychological/neuro characteristic. I would suggest elaborating both introduction and discussion towards how physiological and biomechanical parameters, such as muscle strength and gait parameters, affect psychological parameters such as fear of falling.  

      2. The introduction needs further elaboration, because it refers to the risk of falling, and how this is affected by muscle strength and cognitive function but not to the fear of falling. Please see also previous comment, which might be helpful on how to prepare the reader more appropriately to the aim of the study, i.e. the association of fear of falling with other parameters/characteristics.

Specific comments:

Line 54. Please change "On the contrary" into "Furthermore", since the two sentences do not contradict each other.

Lines 65-67: Please rephrase to become more clear to the reader the meaning of the sentence.

Line 229: Are the values in the table, mean values and standard deviation? Please add this information. Please provide in table 1 the units also for Step Time, Acceleration and Distance.

Table 2. Please rephrase to become more clear to the reader, what is analyzed. Do the authors mean the analyzed parameters?

Line 244: Since there is no p value lower than 0,01, the 2p<0,01 can be deleted.

Line 250-251. Table 3. Please define what are the time reaction parameters. The chair stand test does not evaluate reaction time.

Furthermore, why for SCOWA and FCOWA are 7 values, while for the Chair stand only 6 values. On the table only the 5 out of the 7or 6 values are named. Please add the names of all the values. Is it possible that the Standard error of the β value, and the lower and upper CI are the values names that are missing?

Line 256: I would suggest changing “gait speed” into “gait parameters”.

Line 274-275: It has been reported that a stride length shorter than half of the height is a sensitive risk marker for functional loss and falls (Rodríguez-Molinero, A., Herrero-Larrea, A., Miñarro, A. et al. The spatial parameters of gait and their association with falls, functional decline and death in older adults: a prospective study. Sci Rep 9, 8813 (2019). https://doi.org/10.1038/s41598-019-45113-2). Furthermore, it has been reported that slower walking speed leads to increased stability, irrespective of age (Kang HG, Dingwell JB 2008. Effects of walking speed, strength and range of motion on gait stability in healthy older adults. J Biomech 41(14):2899–2905, doi:10.1016/j.jbiomech.2008.08.002) and that when older and young people walk at self-selected velocity, older adults walk slower and demonstrate higher stability state (Bohm S, Mersmann F, Bierbaum S, Dietrich R, Arampatzis A 2012. Cognitive demand and predictive adaptational responses in dynamic stability control. J Biomech 45(14):2330–2336. doi:10.1016/j.jbiomech.2012.07.009). An explanation for this paradox of higher stability state of the older adults but less safe locomotion (since older adults are more prone to falls during walking than young adults) could be that older adults do not walk slowly enough in relation to their maximum walking velocity, resulting to a lower safety factor during normal locomotion (Mademli L, Arampatzis A. Lower safety factor for old adults during walking at preferred velocity. Age (Dordr). 2014 Jun;36(3):9636. doi: 10.1007/s11357-014-9636-1). Consequently, the statement that longer step lengths and slower gait speeds have been shown to be directly associated with an increased fall risk needs to be further elaborated.

Author Response

All responses to comments are detailed within the attached pdf document.

Reviewer 2 Report

I start by congratulating you on your work. Below I list a series of considerations that we believe are appropriate:

- The abstract does not meet the standards of the journal, we recommend reviewing the format. The authors use acronyms without previously specifying what they refer to. The denomination of measurement tools appears in which it is considered appropriate to name the manufacturer that registers them, they do not provide enough information about the manufacturer and the registration.

We suggest you use mesh terms when selecting keywords

- no objection introduction

-In figure 1 they speak of screened women n=132 and yet the study was not carried out only in women

- The study is descriptive cross-sectional, the same authors speak of an initial sample of 179 subjects but finally they are left with 115. We consider that it is a small sample for the study, we think that it is necessary to implement in 20 -30% of the sample to correct possible losses that occurred in the study.

-In Table 1. Descriptive data of the sample (n = 115), change the commas and put points

-In table 1, can you clarify the term value? Do you mean mean and standard deviation? It is recommended to incorporate the measures of central tendency and deviation corresponding to Table 1.

-We recommend incorporating the P value in another band in Table 2 (P-value)

-At the end of line 261, in the discussion it is considered convenient to start with the results obtained in the study, put a period and continue with the other authors.

- We did not find the permission of the ethics committee to carry out the study.

Author Response

(The authors gave the same response as above.)

Reviewer 3 Report

This an interesting paper especially since it examines the role of fear may play among aging populations regarding  falls .

It is well written but as the authors stated it has limitations that make it in my opinion clinically speaking less useful. Fear is a complex variable and has been identified in other clinical scenarios, like cardiovascular diseases , cancer etc. It would be useful to discuss more the fear and why it was chosen in this study.

The other important clarification that need to be made is that the test chosen for cognitive function is really of very low specificity and no other measure of cognitive function is mentioned although patient with advance dementia were not included. 

Author Response

(The authors gave the same response as above.)

Round 2

Reviewer 1 Report

The authors did a comprehensive revision and have adequately addressed my queries. I only have one last comment.

Line 556-563: It is not so easy to follow the rationale of this part of discussion. I would suggest the part “However, this seems to ……… during normal locomotion [37] to change into something like “In line with this, it has been reported that older adults prefer to walk more slowly than young ones (Hamacher D, Singh NB, Van Dieen JH, Heller MO, Taylor WR (2011) Kinematic measures for assessing gait stability in elderly individuals: a systematic review. J R Soc Interface 8(65):16821698. doi:10.1098/rsif.2011.0416; Alexander NB (1996) Gait disorders in older adults. J Am Geriatr Soc 44(4):434451) because they feel safer this way (Maki BE (1997) Gait changes in older adults: predictors of falls or indicators of fear. J Am Geriatr Soc 45(3):313320) and indeed slower walking speed leads to increased stability, irrespective of age (Kang HG, Dingwell JB (2008) Effects of walking speed, strength and range of motion on gait stability in healthy older adults. J Biomech 41(14):28992905, doi:10.1016/j.jbiomech.2008. 08.002). However, it seems that older adults do not walk slowly enough in relation to their maximum walking velocity, resulting to a lower safety factor during normal locomotion which can explain their less safe locomotion (i.e. older adults are more prone to falls during walking than young adults) (Mademli L, Arampatzis A (2014) Lower safety factor for old adults during walking at preferred velocity. Age (Dordr). Jun;36(3):9636. doi: 10.1007/s11357-014-9636-1).

Reviewer 2 Report

Dear authors,

Once I have reviewed the article, I have to make a series of comments:

- The abstract does not yet meet the publication standards of the journal

- In Table 1, only the substitution of dots for commas has been taken into account, however, the measures of central tendency have not been included.

We request this data because we consider it very important and it has not been provided.

- Finally and fundamentally, taking into account that the study is descriptive cross-sectional, 115 subjects, it is considered an insufficient sample size. From here we encourage you to increase the sample and thus generate more relevant results.

Thank you very much